# Does Geometric Algebra Provide a Loophole to Bell’s Theorem?

**DOI:** 10.3390/e22010061

**Published:** 2019-12-31

**Authors:** Richard David Gill

**Affiliations:** Mathematical Institute, Faculty of Science, Leiden University, Rapenburg 70, 2311 EZ Leiden, The Netherlands; gill@math.leidenuniv.nl

**Keywords:** Bell’s theorem, geometric algebra, Clifford algebra, quantum information

## Abstract

In 2007, and in a series of later papers, Joy Christian claimed to refute Bell’s theorem, presenting an alleged local realistic model of the singlet correlations using techniques from geometric algebra (GA). Several authors published papers refuting his claims, and Christian’s ideas did not gain acceptance. However, he recently succeeded in publishing yet more ambitious and complex versions of his theory in fairly mainstream journals. How could this be? The mathematics and logic of Bell’s theorem is simple and transparent and has been intensely studied and debated for over 50 years. Christian claims to have a mathematical counterexample to a purely mathematical theorem. Each new version of Christian’s model used new devices to circumvent Bell’s theorem or depended on a new way to misunderstand Bell’s work. These devices and misinterpretations are in common use by other Bell critics, so it useful to identify and name them. I hope that this paper can serve as a useful resource to those who need to evaluate new “disproofs of Bell’s theorem”. Christian’s fundamental idea is simple and quite original: he gives a probabilistic interpretation of the fundamental GA equation a·b=(ab+ba)/2. After that, ambiguous notation and technical complexity allows sign errors to be hidden from sight, and new mathematical errors can be introduced.

## 1. Introduction

In this introductory Section 1 will present an overview of the paper and summarise some main themes. The paper discusses a series of papers by J.J. Christian which claims to overthrow our present understanding of quantum information theory by providing a local realistic model of the famous singlet correlations, also known as EPR-B correlations, thereby disproving Bell’s (1964) [1] theorem concerning the incompatibility of quantum mechanics and local realism. Christian resolves the Einstein–Bohr debates (whose highlight was the paper by Einstein, Podolski and Rosen (1935) [2], in favour of Einstein and claims that quantum correlations are an expression of the true geometry of the space-time continuum of the universe. Christian’s main tool is geometric algebra, which may not be familiar to many readers, so I will first summarise the facts we need to know, which are actually no surprise to those coming from quantum information theory, who already know about the geometry of the basic model for one qubit, the Bloch sphere, linking the Hilbert space formalism of quantum mechanics to ordinary 3D geometry.

Recall that Bohm and Aharonov (1957) [3], in order to bring the EPR discussion closer to experimental investigation, had recast Einstein, Podolsky and Rosen’s example (position and momentum of an entangled pair of particles) into the EPR-B example of the horizontal and vertical spins of a pair of spin-half particles in the singlet state, and from there translated it first into the example of the horizontal and vertical polarizations of a pair of photons, and from there into scattering experiments which had been performed recently by Wu and Shaknov (1950) [4].

Of course, this author’s opinion is that Christian’s approach is fundamentally flawed, and throws no light whatsoever on the foundations of quantum mechanics. Moreover, Christian’s “model” went through a number of major transformations, and one can better talk about Christian 1.0, Christian 2.0, and Christian 3.0. With each new edition, more complexity was added, the mistakes of the previous edition were hidden more deeply, new errors were made, new misconceptions were exposed. Since numerous authors criticise Bell’s theorem on the grounds of the same misconceptions, and more generally, follow similar tactics in their attempt to rescue Einstein from Bohr’s victory, I think it is useful to look at this whole process. The work does throw light on how Bell is repeatedly misunderstood.

To warm up, I will recall in Section 2 some basic facts about the set of 2×2 complex matrices $M_{2}\left(C\right)$, thinking of this set as an eight dimensional real vector space endowed with a compatible multiplication operation, namely ordinary matrix multiplication. With these properties, “the algebra $M_{2}\left(C\right)$ over R” is the standard example of the standard geometric algebra of 3D space, also known as the Clifford algebra $C\ell_{3,0}\left(R\right)$. We will meet the famous “GA equation”
(1)a·b=(ab+ba)/2
in the form
(2)$$\left(a\cdot b\right)\;I_{2}\;=\;\left(\left(a\cdot \sigma \right)\left(b\cdot \sigma \right)+\left(b\cdot \sigma \right)\left(a\cdot \sigma \right)\right)/2$$
and connect it to the basic facts of the singlet correlations, in which some standard Quantum Information theory, and in particular, the familiar mathematics and geometry of the Bloch sphere, turns up. In the GA equation, *a* and *b* are ordinary 3D real vectors, with scalar product a·b. The “product” ab is the geometric product and will be defined soon, and we will also make sense of the addition and multiplication of “geometric” objects of various different types. In the $M_{2}\left(C\right)$ version of the equation, $I_{2}$ is the 2×2 identity matrix and σ is the vector of the three Pauli spin matrices.

The GA equation is actually the core of Christian’s opus, a book [5] and very many papers, of which I will discuss seven [6,7,8,9,10,11,12]. He implicitly rewrites the equation as
(3)$$a\cdot b\;=\;\frac{1}{2}ab+\frac{1}{2}ba$$
and interprets this as the expectation value of the random product I{λ=+1}ab+I{λ=−1}ba where I{…} stands for the indicator variable of the named event, and where λ, Christian’s local hidden variable, is a fair coin toss or Rademacher variable—a random variable taking on the values ±1 each with equal probability. Thus he actually considers two geometric products, the usual one, and the transposed one. His hidden variable λ chooses which of the two products to use. His great discovery is to rewrite the GA equation as
(4)$$a\cdot b=E\left(\;a\;\underset{\lambda }{*}\;b\;\right)$$
where I use the mathematician’s notation for expectation value, and where the random binary operation “$\underset{\lambda }{*}$” stands, according to the sign of λ, for the geometric product or its transpose.

Note that the famous negative cosine curve of the singlet correlations, i.e., the correlations between spin measurements in directions *a* and *b* on two entangled spin-half particles in the singlet state, is just the negative cosine of the angle between the two directions, i.e., −a·b. Christian claims that his discovery provides a counter-example to Bell’s theorem [1] on the incompatibility of quantum mechanics and local realism. Christian furthermore claims that these equations are an expression of the torsion of the space in which we live, which, according to him, is the unit sphere in four dimensions $S^{3}$, and he links this idea to the Hopf fibration of $S^{3}$, but I am unable to make any sense of that. Perhaps someone can enlighten me.

Before going into Christian’s works in more detail, I will summarise, in Section 3, some basic results about Clifford algebra (CA) and geometric algebra (GA) in general. In Section 4, I will recall some basic facts about Bell’s inequality and Bell’s theorem, also establishing the terminology which I will use in this paper (many authors mean different things by these two phrases).

As another brief interlude, in Section 5, I will discuss Doran and Lasenby’s (2003) attempt [13] to rewrite the mathematics of a number of interacting qubits in terms of geometric algebra. They succeeded, but their solution is clumsy, and did not catch on. In particular, despite their explicit hope, it did not lead to new interpretations of quantum mechanics.

In Section 6 I will turn at last to a discussion of a chronological sequence of seven selected papers by Christian, focussing on the central ideas and placing them in the general context of opposition to Bell’s theorem.

I will conclude the paper in Section 7 with some ideas about methods and the psychology of Bell-denial in general, and the sociology of science. Of course, I am not a psychologist or a sociologist, nor a physicist, nor a philosopher, so this part of the paper should certainly be taken with many grains of salt. I have the point of view of a mathematician who has specialized in statistics and probability, missing a great deal of mathematical physics in his training, and come relatively late in life to quantum information. I do think that Bell-denial has a great deal to do with two problems in physics. One is that physicists use mathematics as a language in which they speak about the physical world. This makes it harder for them to distinguish between models and reality. Mathematicians do not only calculate; they also study mathematical models, thought of as stand-alone abstract worlds. The common distinction between theoretical physics (part of physics) and mathematical physics (part of mathematics) corresponds, in my mind, to this same difference between on the one hand talking with mathematics about the real world and on the other hand talking with logic about imaginary or toy mathematical worlds. The second problem is the lack of appreciation of many old school physicists for modern probability theory, and lack of knowledge about modern statistics.

By the way, I personally have benefitted enormously from interaction with Bell-deniers, and in particular from a ten year interaction with Joy Christian himself. We have in common a tenacity and conviction that we are right which is probably not healthy. Several Bell-deniers, whom I enormously respect, actually laid their finger on very sore points concerning past experiments, and on very sore points in the common ways that experimental data were analysed. In fact, up till 2015, no experiment had really proved anything. They had, of course, always confirmed quantum mechanics, but that could hardly be a surprise. They had not disproved local realism. It was the unrelenting criticism of a small number of intelligent Bell deniers, in the face of ostracism from the physics establishment, with all the serious effects which this has on their academic careers or scientific reputation, which did actually force the experimental refinements which led up to the first “loophole-free” experiments. The success of these experiments is vital to the reliability of quantum technology which is already being marketed successfully.

The two problems I mentioned are historical problems, since science is evolving, and new generations see scientific hierarchies and divisions differently from old generations. Hopefully these problems will go away, but I fear that new sorts of problems will arise. New dogmas and misconceptions and fashions will take over, for instance, concerning the power of AI and the power of analysis of unscientifically collected “big data”.

Of course there remains the big problem of interpretation of quantum theory, itself connected to the much older problem of the interpretation of probability. These problems are not addressed in this paper, even though Christian believed that he was making a revolutionary contribution to the foundations of quantum theory.

## 2. The Algebra of the 2×2 Complex Matrices over the Reals

An algebra over a field is a vector space over that field endowed with a bilinear product.

Our primary example is the set of 2×2 complex matrices, often denoted $M_{2}\left(C\right)$. One can add them, and one can multiply them by real numbers. This makes $M_{2}\left(C\right)$ into a real vector space.

Every 2×2 complex matrix is a real linear combination of the four matrices with a 1 in one position and three 0s in the other three positions, and the four matrices with an $i=\sqrt{(}-1)$ in one position and three 0s elsewhere. That linear combination equals the zero matrix if and only if all eight coefficients are zero. So as a real vector space, $M_{2}\left(C\right)$ has dimension 8, which makes it isomorphic to $R^{8}$.

We can also multiply matrices. The matrix product is bilinear, that is, it is linear in each argument. Matrix multiplication is associative. We already mentioned the zero matrix; there is also a multiplicative unit, the 2×2 identity matrix. This means that we can think of our space $M_{2}\left(C\right)$ as a real unital associative algebra, or just a real algebra for short. Unital means: with a multiplicative unit. Associativity is such a fundamental property that lack of it is deeply upsetting, and we tend to think of it as an essential part of the concept of algebra. Real means that the scalars in vector-space scalar multiplication are reals.

For now a side remark, it must be mentioned that it is sometimes useful to discuss more general abstract algebras: algebras which are not necessarily associative and not necessarily unital. In particular, we will later briefly meet the octonions, which, oh horror, are not associative. Together with the real numbers, the complex numbers, and the quaternions, these four spaces are sometimes called number systems and denoted R, C, H and O; the letter ‘H’ is for Hamilton. The phrase number system is not a formal mathematical term. The four objects I just listed do have a great deal in common and are in fact the only division algebras. We will come back to them, briefly, later, when discussing one of Christian’s most recent papers.

Let me return to the example of our 8-dimensional real algebra $M_{2}\left(C\right)$. I will now specify an alternative vector-space basis of our space. Define the Pauli spin matrices
(5)$$\sigma_{x}=\left(\begin{array}{cc} 0 & 1\\ 1 & 0 \end{array}\right),\;\;\sigma_{y}=\left(\begin{array}{cc} 0 & -i\\ i & 0 \end{array}\right),\;\;\sigma_{z}=\left(\begin{array}{cc} 1 & 0\\ 0 & -1 \end{array}\right)$$
and the 2×2 identity
(6)$$I_{2}=\left(\begin{array}{cc} 1 & 0\\ 0 & 1 \end{array}\right)$$
and define
(7)$$1=I_{2},\;e_{1}=\sigma_{x},\;e_{2}=\sigma_{y},\;e_{3}=\sigma_{z},\;\beta_{1}=i\sigma_{x},\;\beta_{2}=i\sigma_{y},\;\beta_{3}=i\sigma_{z},\;M=iI_{2}.$$
It is not difficult to check that this eight-tuple is also a vector space basis. Note the following:(8)$$e_{1}^{2}=e_{2}^{2}=e_{3}^{2}=1,$$
(9)$$e_{1}e_{2}=-e_{2}e_{1}=-\beta_{3},\;\;e_{2}e_{3}=-e_{3}e_{2}=-\beta_{2},\;\;e_{3}e_{1}=-e_{1}e_{3}=-\beta_{1},$$
(10)$$e_{1}e_{2}e_{3}=M.$$
Denote the zero matrix by 0. These relations show us that the 2×2 complex matrices, as an algebra over the reals, are isomorphic to the Clifford algebra (CA) $C\ell_{3,0}\left(R\right)$. In other words, they form a representation, or concrete realisation, of this algebra. By definition, $C\ell_{3,0}\left(R\right)$ is the associative unital algebra over the reals generated from 1, $e_{1}$, $e_{2}$ and $e_{3}$, with three of $e_{1}$, $e_{2}$ and $e_{3}$ squaring to 1 and none of them squaring to −1. This is the meaning of the “3” and the “0” in the notation: three squares are positive, zero are negative. Moreover, the three generating elements $e_{1}$, $e_{2}$ and $e_{3}$ anti-commute. The smallest possible algebra over the real numbers which can be created with these rules has vector space dimension $2^{3+0}=1+3+3+1=8$ and as a real vector space, a basis of 8 linearly independent elements can be taken to be 1, $e_{1}$, $e_{2}$, $e_{3}$, $e_{1}e_{2}=-\beta_{3}$, $e_{1}e_{3}=\beta_{2}$, $e_{2}e_{3}=-\beta_{1}$, and $e_{1}e_{2}e_{3}=M$. The algebra is associative (like all Clifford algebras), but not commutative.

## 3. Clifford Algebras over the Reals

My description was not quite the “official” definition of a Clifford algebra, but is sufficient for our purposes. We are mainly interested in the 8-dimensional algebra $C\ell_{3,0}\left(R\right)$, which is often called the Clifford algebra of 3D real geometry. It is also called the (or a) geometric algebra and the product is called the (or a) geometric product.

A bit more generally, let *p* and *q* be nonnegative integers. The Clifford algebra $C\ell_{p,q}\left(R\right)$ is the real algebra of dimension $2^{p+q}$ generated by p+q anti-commuting elements $e_{i}$, of which *p* square to +1 and *q* to −1. The multiplicative unit 1 commutes with all elements of the algebra. Scalar multiples of 1 can be identified with the scalar itself. In particular, we can safely write 1=1, 0=0. Omitting the (in this paper) omnipresent field (R), one easily sees that $C\ell_{0,0}=R$, $C\ell_{0,1}=C$. A bit less obviously, $C\ell_{0,2}=H$, the quaternions.

Each Clifford algebra has an even sub-algebra, which by definition is the smallest algebra containing all multiples of an even number of the generating elements $e_{i}$. As an example, one can check that $C\ell_{3,0}^{=}\;=H$, the quaternions again.

Thus within our favourite algebra $C\ell_{3,0}$, many famous and familiar structures are embedded.

We already mentioned that its even subagebra can be identified with the quaternions. This is the real linear algebra of linear combinations of the four elements $1,\beta_{1},\beta_{2},\beta_{3}$. Notice that
(11)$$\beta_{1}^{2}=\beta_{2}^{2}=\beta_{3}^{2}=-1,\;$$
(12)$$\beta_{1}\beta_{2}=-\beta_{2}\beta_{1}=-\beta_{3},\;$$
(13)$$\beta_{2}\beta_{3}=-\beta_{3}\beta_{2}=-\beta_{1},\;$$
(14)$$\beta_{3}\beta_{1}=-\beta_{1}\beta_{3}=-\beta_{2}.\;$$
This is close to the conventional multiplication table of the quaternionic roots of minus one: the only thing that is wrong is the last minus sign in each of the last three rows. However, the sign can be fixed in many ways; for instance, by taking an odd permutation of $\left(\beta_{1},\beta_{2},\beta_{3}\right)$, or an odd number of sign changes of elements of $\left(\beta_{1},\beta_{2},\beta_{3}\right)$. In particular, changing the signs of all three does the job. In particular, the triple $e_{1}e_{2}=-\beta_{3},e_{1}e_{3}=\beta_{2},e_{2}e_{3}=-\beta_{1}$ (the order is crucial) does the job. We reversed the order of the three $\beta_{j}$s, which is an odd permutation, and changed two signs, an even number. We ended up with, in order, $e_{1}e_{2}$, $e_{1}e_{3}$, and $e_{2}e_{3}$. In fact this is also the the ordinary lexicographic ordering of the three products.

It must be emphasised that switching some signs or permuting the order does not change the algebra generated by $1,\beta_{1},\beta_{2},\beta_{3}$, though Christian seems to think this is the case. Choosing a different vector space basis does not change the vector space as a set, nor the operations of addition, multiplication and scalar multiplication. The definition of a Clifford algebra does not entail any particular choice of order of the elements of any particular vector space basis. It is possible to work with different bases of a vector space at the same time. If we do want to do that, we must make it clear either by notation or by context which basis is being used at any given time. Not doing this, is Christian’s main device for introducing sign changes in order to get the results he wants, as I already mentioned in the introduction to the paper.

Obviously, *M* commutes with everything, and obviously $M^{2}=-1$. Also $Me_{1}=\beta_{1}$, $Me_{2}=\beta_{2}$, $Me_{3}=\beta_{3}$ and, in duality with this, $e_{1}=-M\beta_{1}$, $e_{2}=-M\beta_{2}$, $e_{3}=-M\beta_{3}$. Thus any element of $C\ell_{3,0}\left(R\right)$ can be expressed in the form p+Mq where *p* and *q* are quaternions, *M* commutes with the quaternions, and *M* is itself another square root of minus one.

Notice that $\left(1-e_{i}\right)\left(1+e_{i}\right)=0$. So the algebra $C\ell_{3,0}\left(R\right)$ possesses zero divisors (in fact, very many!), and hence not all of its elements have inverses.

The real reason for calling this algebra a geometric algebra comes from locating real 3D space within it, and also recognising geometric operations and further geometric structures with the algebra. To start with, look at the linear span of $e_{1}$, $e_{2}$ and $e_{3}$. This can be identified with $R^{3}$: from now on, real 3D vectors are real linear combinations of $e_{1}$, $e_{2}$ and $e_{3}$. So let us look at two elements *a*, *b* of $R^{3}$, thus simultaneously elements of the linear span of $e_{1}$, $e_{2}$ and $e_{3}$.

In the latter incarnation (i.e., as elements of $C\ell_{3,0}\left(R\right)$), we can multiply them; what do we find? The answer is easily seen to be the following:(15)ab=(a·b)1+M(a×b).

Here, a·b stands for the (ordinary) inner product of two real (three dimensional) vectors, hence a real number, or a scalar; a×b stands for the (ordinary) cross product of two real three dimensional vectors, hence a real vector. As such, it is a real linear combination of $e_{1}$, $e_{2}$ and $e_{3}$. Multiplying by *M* gives us the same real linear combination of $\beta_{1}$, $\beta_{2}$, $\beta_{3}$.

Note that we immediately find from this relation that
(16)$$a\cdot b=\frac{1}{2}\left(ab+ba\right).\;$$
This is the magic formula which Christian has taken as the basis for his local hidden variables model of the singlet correlations. Multiply throughout by −1 and interpret 1/2 as a probability.

Thus the geometric algebra $C\ell_{3,0}\left(R\right)$ contains as a linear subspace the real vectors of $R^{3}$; the Clifford algebra product, which we now call the geometric product, of two such elements encodes both their scalar dot product and their vector cross product. The dot product and the cross product of real vectors can both be recovered from the geometric product, since these parts of the geometric product live in parts of the eight dimensional real linear space $C\ell_{3,0}\left(R\right)$ spanned by disjoint sets of linearly independent basis elements.

The real linear subspace generated by $\beta_{1}$, $\beta_{2}$, $\beta_{3}$ is called the set of bivectors, just as the real linear subspace spanned by $e_{1}$, $e_{2}$, $e_{3}$ is identified with the set of vectors. As we noted before, the real linear subspace spanned by the single element 1 is identified with the real numbers or scalars and in particular, we identify the scalar 1 and the element 1, the scalar 0 and the element 0. The real linear subspace generated by the single element *M* is called the set of trivectors, also known as pseudo-scalars.

Bivectors can be thought of as oriented planes; trivectors as signed volume elements. Going further still, rotation of vectors can be represented in terms of bivectors and so-called rotors.

So we have seen that $C\ell_{3,0}\left(R\right)$ is a beautiful object, containing within itself the quaternions, the complex numbers, the 2×2 complex matrices, three dimensional real vectors; and its product, called “geometric product”, contains within it both the inner and outer product of real vectors. Everything comes in dual pairs: the vectors and the bivectors; the quaternions and the product of the quaternions with *M*, the scalars and the pseudo-scalars. Every element of the algebra can be split into four components: scalar, vector, bivector and pseudo-scalar. The algebra is called graded. The parts we just named constitute the zeroth, first, second and third grades of the algebra. The product of elements of grades *r* and *s* belong to the grade r+s modulo 4.

The fact that $C\ell_{3,0}\left(R\right)$ has as a representation the algebra of complex 2×2 matrices over the reals is, on the whole, not much more than a side-note in the literature on geometric algebra and on Clifford algebra; it is just one tiny piece in the theory of classification of Clifford algebras in general. It seems to this author that this representation should help very much to “anchor” the theory for those coming from quantum theory and quantum foundations. The standard textbook on geometric algebra for physicists, Doran and Lasenby (2003) [13], has two whole chapters on quantum mechanics, the first containing a geometric algebraic representation of the quantum theory of one spin half particle, the second doing the same for two entangled spin half particles. This even contains a carefully worked out analysis of the singlet state, including derivation of the EPR-B correlation −cos(a·b).

The reader is referred to Chapters 8 and 9 of that work whose endnotes contain many, many further references. What is important to notice for the main theme of the present paper, is that prior to Christian’s work, there already was a large literature devoted to expressing quantum theory in terms of geometric algebra. Though it seems that this literature did not gain a place in mainstream quantum theory, one of Christian’s aims had already been fulfilled: geometric algebra had already been put forward as a vehicle for re-writing quantum theory with geometry instead of Hilbert space at the forefront. I will later try to explain why, I believe, Doran and Lasenby’s project (following the foundation laid by David Hestenes, see [14,15]) to geometrise quantum information theory never caught on.

### Remarks on Computation

Before continuing, I would like to make some remarks on computation. In order to test formulas, or in order to simulate models, it is convenient to have access to computer languages which can “do” computer algebra.

First of all, the fact that $C\ell_{3,0}\left(R\right)$ can be identified with the two-by-two complex matrices means that one can implement geometric algebra as soon as one can “do” complex matrices. One needs to figure out how to get the eight coordinates out of the matrix. There are easy formulas but the result would be clumsy and involve a lot of programming at a low level. The relation with the quaternions is more promising: we can represent any element of $C\ell_{3,0}\left(R\right)$ as a pair of quaternions, and the real and the imaginary parts of the two component quaternions immediately give us access to the scalar and bivector, respectively trivector (also called “pseudo scalar”) and vector parts of the element of the algebra.

For those who like to programme in the statistical language R it is good to know that there is an R package called “onion” which implements both the quaternions and the octonions.

Nicest of all is to use a computer system for doing real geometric algebra, and for this purpose I highly recommend the programme *GAViewer* which accompanies the book Dorst, Fontijne and Mann (2007) [16]. It can be obtained from the authors’ own book website http://www.geometricalgebra.net. It not only does geometric algebra computations, it also visualises them, i.e., connects to the associated geometry. Moreover the book and the programme are a nice starting point for two higher dimensional geometric algebras, of dimension 16 and 32 respectively, which encode more geometric objects (for instance, circles and affine subspaces), and more geometric operations, with the help of the further dimensions of the bigger algebras. “ We will have more to say about some other Clifford algebras, and also about the octonions (which are not a Clifford algebra), later. For the moment, we just make a few remarks. The octonions are an eight dimensional algebra which is not associative. There are wonderful and deep connections between these various objects and in particular, concerning the sequence 1, 2, 4, 8, which is where a particular construction taking us from the reals to the complex numbers to the quaternions and finally to the octonions stops. The octonions can be thought of as ordered pairs of quaternions, which can be thought of as ordered pairs of complex numbers, which can be thought of as ordered pairs of real numbers. These four algebras are the only division algebras: algebras where every non-zero element has an inverse. Equivalently, there are no zero divisors (zero cannot be written as a product of non-zero elements) except zero itself.

Another Clifford algebra of dimension 8, $C\ell_{0,3}\left(R\right)$, can also be thought of as ordered pairs of quaternions.

So far I have mentioned two (now classic) text books both written to promote geometric algebra, one in physics, the other in computer science. A recent populariser of geometric algebra is Derek Abbott, who together with his colleagues, has also used it in quantum information theory, namely in quantum game theory. I highly recommend the group’s recent historical survey paper, Chappell et al. (2016) [17].

## 4. Bell’s Theorem

The phrase “Bell’s theorem” means different things to different authors, even to the originator himself in different phases of his career. I would like to distinguish Bell’s inequalities, or more generally, Bell-type inequalities, from a theorem which I will call Bell’s theorem, which asserts the incompatibility of quantum theory and local realism. In this terminology, Bell’s inequality (both the original three correlation inequality and the later Bell-CHSH four correlations inequality) are merely simple lemmas using some elementary probability theory, which is applicable once we have assumed the existence of a local hidden variables model. Both the original 1964 Bell inequality, and the more general CHSH inequality [18] of Clauser, Horne, Shimony and Holt (1969), which was then on always promoted by Bell himself, can be seen as an application of the elementary Boole inequality, stating that the probability of a union of events is less than or equal to the sum of their separate probabilities. Amusingly, Boole himself used the Boole inequality to derive Bell’s three correlation inequality, as an exercise to the reader in his magnum opus (1854) book on logic and probability theory [19]. He did not provide a solution manual! Some Bell critics have accused Bell of stealing Boole’s result without proper reference to the source. This is a ridiculous accusation, in my opinion, which merely shows how ignorant some top scientists can be of elementary probability theory. I wonder who else has read Boole’s book recently from cover to cover?

Bell’s theorem is usually proved by the following steps. Assuming local realism, derive the (“four correlations”) Bell-CHSH inequality. Show that a particular choice of quantum mechanical state and particular choice of measurements gives a violation of the inequality. The original Bell (1964) three correlations inequality is a special case of CHSH, which follows on assuming that one of the correlations is identically equal to minus one. Bell himself in 1964 already realised that if one wished to apply his theory to experimental results, one would have to take account of the fact that any real experiment is probably not going to prove that the correlation which was hoped to be a perfect minus one, is actually not quite perfect. Experimental results would result in a failure of the experimenter’s aim. Bell went on to show that there was some room for experimental error, in particular, that one did not have to prove perfect correlation or anti-correlation, but only to some close enough approximation. Thus he already envisaged experimentally investigating four correlations, not three, and he was hardly surprised when CHSH came along, and espoused it straight away.

But this is not the only way to prove Bell’s theorem, and of course, actually we can prove much more. One can for instance show that the collection of probability distributions of joint measurement outcomes in a standard Bell-type experiment (two parties, two settings per party, two outcomes per setting per party) under local realism is strictly contained in the collection of probability distributions under quantum mechanics. In experiments, we need not aim at the particular state and particular settings which Bell exhibited, which later (with the Tsirelson inequality of 1980, [20]) turned out to actually be the very best that quantum mechanics can do, with regards to the numerical size of the violation of the CHSH inequality. The collection of CHSH inequalities obtained by permuting outcomes per setting per party, permuting settings per party, and permuting parties, turned out not only to necessarily hold under local realism, but that the converse is also true: if all eight hold, together with the no-signalling and normalisation constraints which we know must hold anyway both in reality and in any decent model of reality, then there is a local realist model which explains the “observed” joint probability distributions exactly; this result is due to Fine (1982) [21]. The hidden variable model in question is a probability mixture of one of just sixteen “extreme” local hidden variables models, namely models in which all probabilities are either zero or one. In other words, the local hidden variable can be taken to be discrete, just taking on 16 different possible values. The probability distribution of the hidden variable is specified by fixing 16 probabilities which add to one.

One can also consider scenarios with more than two settings per party. Various somewhat more simple proofs of Bell’s theorem, for instance, one due to Mermin, involves three settings per party, so a total of nine component sub-experiments. There are proofs of Bell’s theorem “without inequalities”, for instance one due to Hardy, who found a state and measurements such that a certain event which should have probability zero under local realism actually had positive probability under quantum mechanics. It later turned out that this state and measurements were useful in designing experiments which were less sensitive to noise and imperfections. In a sense, it turned out that less entanglement could actually go together with more quantum non-locality.

In my opinion, these more or less standard proofs of Bell’s theorem are examples of theorems from computer science, and in particular, in the subfield called distributed computing. One part of the theorem uses a simple lemma (Bell’s inequality) from classical distributed computing; the other part (the violation by QM) is a simple lemma from quantum distributed computing.

Other proofs are inspired by functional analysis, or if you prefer, numerical approximation theory. The idea is to ask the question: can the whole correlation function of spin measurements on two spin systems in the singlet state be approximated in the way that local realism suggests, namely as an integral over the possible values of a hidden variable of two “measurement functions”, both depending on the hidden variable, but otherwise, each depending on just one of the two settings (spin measurement directions). Thus we have a known function of two spatial directions and *a* and *b*, namely the negative cosine of the angle between those two directions, and ask if it can be written as $\int A\left(a,\lambda \right)B\left(b,\lambda \right)P_{\rho }\left(d\lambda \right)$, where $P_{\rho }$ is a fixed probability measure (often supposed to have a probability densithy ρ with respect to some dominating measure); the functions *A* and *B* take the values ±1. This does correspond to early experiments (usually involving polarization of photons, not spin of electrons) in which polarizing beam-splitter directions were more or less continually scanned in both wings of the experiment in an attempt to reproduce the whole correlation function, not just four points on it. Steve Gull in particular gave a beautiful impossibility proof using Fourier analysis in slides of a talk which can be found on his home page at the University of Cambridge, http://www.mrao.cam.ac.uk/~steve/maxent2009/images/bell.pdf. He poses the problem as a computer challenge: write computer programs for two separate computers which will reproduce these correlations … or prove that it cannot be done, and sketches the argument, see my preprint Gill (2013) [22] for more details.

Such a proof would seem to assume some amount of mathematical regularity conditions. Is the integral a Lebesgue integral? Are the functions measurable? It has been argued by Ingmar Pitowsky ([23] (chapter 5), and after that, by many others, who tend not to know Pitowsky’s work!) that non-measurability was a loophole; I find this argument completely spurious. Anyway, there are also proofs of Bell’s theorem which essentially only use discrete probability. One was found by myself, Gill (2014) [24]. Earlier still, a martingale approach also discovered by myself, references [25,26], allows a proof of Bell’s theorem which is “driven” purely by the probability involved in the random binary choice of measurement directions. Thus the part of the proof in which we assume local realism involves truly only the counting of the $4^{N}$ outcomes of tosses of two fair coins: how many lead to a violation of Bell’s inequality which is so large that strong doubt is raised against the hypothesis of local realism. These ideas have been both refined and simplified and used in the famous “loophole free” experiments of 2015.

The point of these remarks is just to say that Bell’s theorem is a rather simple theorem, which can be proved in many different ways using tools from widely different parts of mathematics. In particular, the idea that one could somehow avoid the conclusion of the theorem by supposing that the hidden variable takes values in some weird and wonderful (and little known) mathematical structure, is completely unfounded. We do not need any measurability or other kinds of regularity of those measurement functions. Similarly, the idea that Bell’s theorem is a paradox due to some kind of inadequacy of “standard mathematics” is a fata morgana. Bell’s theorem cannot be “explained” by supposing that in physics we need to use non-standard calculus or generalised notions of differentiation or integration. It cannot be explained by supposing the ZFC axioms (Zermelo-Fraenkel set theory with the axiom of choice) are inconsistent and have to be abandoned. The argument here is that the ZFC axioms lead to a contradiction—Bell’s theorem. The axioms therefore allow one both to prove a certain statement (the inequality) and its negation (violation of the inequality).

Naturally, just occasionally, it does turn out that a theorem which everyone believed was true, turned out to have been wrong. There had been some failure of the imagination. Imre Lakatos [27] has wonderful examples from the history of mathematics: for instance, the inventor of rigour in calculus, Cauchy, had also proved that the pointwise limit of a sequence of continuous functions is also continuous, and for decades everyone both knew the theorem and knew counter-examples, coming from Fourier analysis. The attempt to rigourize Fourier analysis actually led to the invention of both functional analysis and of Riemann integration and later of measure theoretic integration. So it is apparently not impossible that everyone has overlooked some hidden assumption in Bell’s arguments. Every year a few authors come up with their own discovery of such a hidden assumption. Nowadays, thanks to the rise of numerous predatory journals, they even pass peer review and get published. So far, they have always been (in my opinion) wrong, and in any case, it remains a fact that though there is a sizeable community of “Bell-deniers”, none of them agree with one another with what was wrong, and many of them admit that this is the case. But was Joy Christian the exception which proved this rule?

It is clear that since the title of the present paper ends in a question mark, my answer to this question is emphatically no.

## 5. Geometrizing Quantum Information Theory

I assume that the reader is already familiar with the standard “Bloch sphere” picture of quantum states, quantum operations, and quantum measurements for a single qubit. This means that the first of Doran and Lasenby’s two chapters on quantum information in their 2003 textbook [13] hold few surprises. Quantum states can be represented as points in the Bloch ball, thus as real 3D vectors of length less than or equal to one. Pure states lie on the surface of the ball, mixed states in the interior. Unitary evolutions correspond to rotations of the Bloch sphere and these are represented by right and left multiplication with a unitary matrix and its Hermitian conjugate. In geometric algebra, rotation of a 3D real vector is expressed using an object called a “rotor” and the geometric idea is that we do not rotate around a vector, but in a plane. Oriented planes are represented by bivectors.

Incidentally, Doran and Lasenby (2003) are actually reporting on the results they obtained, carrying out the programme already set out by David Hestenes, the pioneer rediscoverer and populariser of geometric algebra as a language for physics, see [14,15]. See Chappell et al. [17] for some of the history of the struggle between two kinds of vector algebra, Gibbs versus Clifford. Incidentally again, one of the authors of that survey, D. Abbott, has also been promoting Christian’s recent works.

One curious point is that Doran and Lasenby also attempt to map the vector (wave function) representation of a pure state into the geometric algebra framework. Since the density matrix is a completely adequate representation of any state, pure or mixed, this seems to me to be superfluous. Why not agree that a pure state is represented by its density matrix, hence a point on the unit sphere, hence a real vector of unit length? They do achieve their aim by thinking of state vectors as being the result of a 3D rotation from a fixed reference direction, hence pure states are represented by rotors which involve a choice of reference direction.

All in all, one could argue that rephrasing the mathematics of a single qubit in terms of real geometric algebra demystifies the usual approach: no more complex numbers, just real 3D geometry. Hestenes [15] already hinted at the possibility, as do Doran and Lasenby several times in their book and other publications, that this approach has the promise of delivering a new and intuitive geometric interpretation of quantum mechanics, but they never appear to follow this up. Perhaps it depends on what is meant by “intuitive” and “interpretation”.

Having all the mathematical machinery of the complex two-by-two matrices at their disposal, it is essentially a straightforward matter to now also extend the discussion to several (entangled) qubits. Indeed, their book contains a reworking of the computation of the singlet correlations for the special case of a maximally entangled pair of qubits.

The approach is simply to take the (outer) product of as many copies of $C\ell_{3,0}\left(R\right)$ as we have qubits. Thus no new geometry is involved and in fact we move out of the realm of Clifford algebras: a tensor product of several Clifford algebras is not a Clifford algebra itself. Certainly it is the case that the outer product of two copies of $C\ell_{3,0}\left(R\right)$ contains, as an algebra, everything that we need relative to the usual Hilbert space approach. However it contains more than we need. The problem is that in the Hilbert space approach, the tensor product of the identity with *i* times the identity is equal to the tensor product of *i* times the identity with the identity. But taking the outer product of two Clifford algebras defines two distinct elements: 1⊗M and M⊗1. Thus we should form the basic space for a composite system not just by taking a product of spaces for the subsystems but also by adding a further equality 1⊗M=M⊗1.

Notice that the 2×2 complex matrices have real dimension eight, while the 4×4 complex matrices have real dimension 32. However, 8×8=64. Thus if we believe the usual Hilbert space story for the mathematics of two qubits, then states, operators, measurements, … can all be represented in a real algebra of dimension 32. Adding an extra equality 1⊗M=M⊗1 reduces the dimension of $C\ell_{3,0}{\left(R\right)}^{⊗2}$ from 64 to 32 and now we do have exactly the algebra which we need in order to carry the “usual” Hilbert space based theory.

In fact, Doran and Lasenby, taking their cue from their representation of pure states as rotors, representing a rotation from a reference direction to the direction of the state, construct a different reduction by imposing a different equality. Their point is that geometrically, we need to align the reference directions of the two subsystems. Perhaps this is physically meaningful too. This leads to an algebraically more complicated “base space”, with perhaps better geometric interpretability than the obvious algebraic reduction, which comes down to identifying the complex root of minus one, *i*, as being “the same” in the two Hilbert spaces of the two subsystems: mathematically that is a natural step to take, but what does it mean physically?

The proof of the pudding should be in the eating. On the one hand, everything one can do with the Clifford algebra approach can also be done in the Hilbert space approach: after fixing the dimension issue, one has two mathematically isomorphic systems, and the choice between the two is merely a matter of “picking a coordinate system”. So far, it seems that the Clifford algebra approach did not pay off in terms of easier computations or geometric insight, when we look at systems of several qubits. On the other hand, the geometric insight for a single qubit was already available and has become completely familiar. The geometric algebra approach did not lead to new interpretational insights in quantum theory. Finally, it does not extend in an easy way to higher-dimensional systems: where is the geometric algebra of the qutrit?

However I hope that this paper will encourage others to take a fresh look for themselves.

## 6. Christian’s Disproofs of Bell’s Theorem

### 6.1. Christian’s First Model

After several pages of general introduction, Christian [6] gives a very brief specification of his (allegedly) local realist model for the singlet (or EPR-B) correlations, obtained through the device of taking the co-domain of Bell’s measurement functions to be elements of the geometric algebra $C\ell_{3,0}\left(R\right)$ rather than the conventional (one dimensional) real line. Note that he refers not to the domain, unlike many other Bell-deniers, who like the hidden variable to live in some really exotic space. No, he refers to the space of outcomes of the measurement function, which Bell took to be the set {−1,+1}. Nowadays, experimenters are also very well aware of this requirement, though occasionally they do miss that point in the raison d’être of their experiment.

Conventionally, a local hidden variables model for the singlet correlations consists of the following ingredients. First of all, there is a hidden variable λ which is an element of some arbitrary space over which there is a probability distribution referred to in physicist’s language sometimes as ρ(λ)dλ, sometimes as dρ(λ). This hidden variable is often thought to reside in the two particles sent to the two measurement devices in the two wings of the experiment, and therefore to come from the source; but one can also think of λ as an assemblage of classical variables in the source and in both particles and in both measurement devices which together determine the outcomes of measurement at the two locations. Any “local stochastic” hidden variables model can also be re-written as a deterministic local hidden variables model. This rewriting (thinking of random variables as simply deterministic functions of some lower level random elements) might not correspond to physical intuition but as a mathematical device it is a legitimate and powerful simplifying agent. Bell himself already knew these tricks and mentioned them. Many Bell critics unfortunately do not have the mathematical sophistication to get the hints which he already laid down in 1964 in [1]. Their argument against Bell’s hidden variable model is that it is unphysical. It looks unphysical, but that is because one is jumping to conclusions, supposing that Bell’s model says that λ comes from the source and from the source alone. No it does not, and Bell already pointed that out!

Secondly, we have two functions A(a,λ) and B(b,λ) which take the values ±1 only, and which denote the measurement outcomes at Alice’s and Bob’s locations, when Alice uses measurement setting *a* and Bob uses measurement setting *b*. Here, *a* and *b* are 3D spatial directions conventionally represented by unit vectors in $R^{3}$. The set of unit vectors is of course also known as the unit sphere $S^{2}$.

Bell’s theorem states that there do not exist functions *A* and *B* and a probability distribution ρ, on any space of possible λ whatever, such that
(17)∫A(a,λ)B(b,λ)dρ(λ)=−a·b
for all *a* and *b* in $S^{2}$.

Christian claims to have a counter-example and the first step in his claim is that Bell “unphysically” restricted the co-domain of the functions *A* and *B* to be the real line. Now this is a curious line to take: we are supposed to assume that *A* and *B* take values in the two-point set {−1,+1}. In fact, the correlation between *A* and *B* in such a context is merely the probability of getting equal (binary) outcomes minus the probability of getting different (binary) outcomes. In other words: Bell’s theorem is about measurements which can only take on two different values, and it is merely by convention that we associate those values with the numbers −1 and +1. We could just as well have called them “spin up” and “spin down”. In the language of probability theory, we can identify λ with the element ω of an underlying probability space, and we have two families of random variables $A_{a}$ and $B_{b}$, taking values in a two point set, without loss of generality the set {−1,+1}, and Bell’s claim is: it is impossible to have $\mathrm{Prob}\left(A_{a}=B_{b}\right)-\mathrm{Prob}\left(A_{a}\neq B_{b}\right)=-a\cdot b$ for all *a* and *b*.

However, let us bear with Christian, and allow that the functions *A* and *B* might just as well be thought of as taking values in a geometric algebra … as long as we insist that they each only take on two different values.

Christian used the symbol n to denote an arbitrary unit vector (element of $S^{2}$) and in formulas (15) and (16) of [6] makes the following bold suggestion:(18)$$A_{n}=B_{n}=\mu \cdot n≅\pm 1\in S^{2}\;$$
where μ·n has been previously defined to be ±Mn (I use the symbol *M* instead of Christian’s *I*). Christian talks about the dot here standing for a “projection” referring, presumably, to taking the lowest grade part, the scalar part, of the GA product. Some more explicit definitions and references would be useful (e.g., references to a formula or page number in a book, not just to a book). Christian sees μ as his local hidden variable, giving us the story that space itself picks at random a “handedness” for the trivector $M=e_{1}e_{2}e_{3}$, thought of as a directed volume element. This story is odd; after all, the “handedness” of $e_{1}e_{2}e_{3}$ is merely the expression of the sign of the evenness of the permutation 123. Of course, the multiplication rule of geometric algebra, bringing up the cross product does again involve a handedness convention: but this is nothing to do with physics, it is only to do with mathematical convention, i.e., with book-keeping.

But anyway, within Christian’s “model” as we have it so far, we can just as well define λ to be a completely random element of {−1,+1}, and then define μ=λM. The resulting probability distribution of μ is the same; we have merely changed some names, without changing the model.

So now we have the model
(19)A(a,λ)=λMa,B(b,λ)=λMb
which says that the two measurement functions have outcomes in the set of pure (unit length) bivectors. Now, those two sets are both isomorphic to $S^{2}$, and that is presumably the meaning intended by Christian when using the congruency symbol ≅: our measurements can be thought of as taking values in $S^{2}$. At the same time, each measurement takes on only one of two different values (given the measurement direction) hence we can also claim congruency with the two point set {−1,+1}={±1}. But of course, these are two different congruencies! Furthermore, they still need to be sorted out. What is mapped to what, exactly …

This is where things go badly wrong. On the one hand, the model is not yet specified, if Christian does not tell us how, exactly, he means to map the set of two possible values ±Ma onto {±1} and how he means to map the set of two possible values ±Mb onto {±1}. On the other hand, Christian proceeds to compute a correlation (in the physicist’s sense: the expectation value of a product) between bivector valued outcomes instead of between {±1} valued outcomes. No model, wrong correlation.

Let us take a look at each mis-step separately. Regarding the first (incomplete specification), there are actually only two options (without introducing completely new elements into the story), since Christian already essentially told us that the two values ±1 of (my) λ are equally likely. Without loss of generality, his model (just for these two measurement directions) becomes either
(20)A(a,λ)=B(b,λ)=λ=±1
or
(21)−A(a,λ)=B(b,λ)=λ=±1
and hence his correlation is either +1 or −1, respectively.

Regarding the second mis-step, Christian proceeds to compute the geometric product (μ·a)(μ·b) (later, he averages this over the possible values of μ). Now as we have seen this is equal to $\left(\lambda Ma\right)\left(\lambda Mb\right)=\lambda^{2}M^{2}\mathrm{ab}=-a\cdot b-M\left(a\times b\right)$ and therefore certainly not equal to −a·b−λM(a×b)=−a·b−μ(a×b).

### 6.2. Christian’s Second Model

I next would like to take the reader to Christian’s “one page paper” Christian (2011) [6], simultaneously the main material of the first chapter of his book Christian (2014) [5].

It seems clear to me that by 2011, Christian himself has realised that his “model” of 2007 was incomplete: there was no explicit definition of the measurement functions. So now he does come up with a model, and the model is astoundingly simple … it is identical to my second model:(22)A(a,λ)=−B(b,λ)=λ=±1.
However, he still needs to get the singlet correlation from this, and for that purpose, he daringly redefines correlation, by noting the following: associated with the unit vectors a and b are the unit bivectors (in my notation) Ma and Mb. As purely imaginary elements of the bivector algebra or quaternions, these are square roots of minus one, and we write
(23)A(a,λ)=(Ma)(−λMa)=λ
(24)B(b,λ)=(λMb)(Mb)=−λ
where λ is a “fair coin toss” or Hadamacher random variable, i.e., equal to ±1 with equal probabilities $\frac{1}{2}$.

Now the cunning device of representing these two random variables as products of fixed bivectors and random bivectors allows Christian to propose a generalised Pearson bivector correlation between *A* and *B* by dividing the mean value of the product by the non random “scale quantities” Ma and Mb. In other words, Ma and Mb are thought of as standard deviations. The Clifford algebra valued measurement outcome ±1 is thought of as having a standard deviation which is a bivectorial root of −1.

Since the geometric product is non commutative, one must be careful about the order of these operations (and whether we divide on the left or the right), but I will follow Christian in taking what seems natural choices: “a” always on the left, “b” on the right.

Unfortunately, since AB=−1 with probability one, the Christian–Pearson correlation should be $-{\left(Ma\right)}^{-1}{\left(Mb\right)}^{-1}=-\left(Ma\right)\left(Mb\right)=-a\cdot b-M\left(a\times b\right)$, just as before. However, just as in the 2007 paper, Christian succeeds again in 2011 in getting a sign wrong, so as to “erase” the unwanted bivector cross-product term from the “correlation”. In Gill (2012) [28] I have analysed where he went wrong and put forward an explanation of how this mistake could occur (ambiguous notation and inaccurate terminology, driven of course by powerful wishful thinking). It should be noted that he also hides the sign error somewhat deeper in complex computations, looking at the average of a large number of realisations and using the law of large numbers, rather than just computing the expectation “theoretically”. I have explained the error also in the introduction to this paper. The key GA equation is interpreted as an expectation value, $a\cdot b=\frac{1}{2}ab+\frac{1}{2}ba$. Thus we are dealing, with probability half, with the usual geometric product, and with probability half, with the transposed geometric product. Apparently nature itself is choosing with a fair coin toss λ which product to use. The two different products should have been notationally distinguished using some kind of annotated product symbol in which λ appeared explicitly, see the Introduction.

Christian’s computer programming supporters discovered themselves that his formulas did not give the right answer, and discovered the “fix”. His GA model is an amusing way to compute a cosine function by a Monte-Carlo simulation, in which the cosine itself is already built in: GA knows the cosine function since it already knows how to compute a·b. The Monte-Carlo part is needed to average out the part of the geometric products involving the cross product or its transpose.

I will now go on to discuss subsequent versions of Christian’s model. He has elaborated more or less the same “theory” with the same repertoire of conceptual and algebraic errors in various papers with increasing levels of complexity. This work shows a remarkable degree of dedication and persistence, and erudition too, as more and more complex mathematical constructions are brought into play. However the flaws in the foundation are unchanged and one of them (as has been said by many before) is unredeemable: Bell’s theorem is about probabilities of different combinations of pairs of binary outcomes. A local hidden variable theory has to explain the correlations between binary outcomes which are predicted by quantum mechanics, following standard, minimal, interpretations: quantum mechanics predicts the probability distribution of measurement outcomes. Experimenters (in this field) count discrete events and estimate probabilities with relative frequencies, and then compute correlations as defined to be the probability of equal outcomes minus the probability of opposite outcomes. A statistician would call them unnormalised and uncentered (or “raw”) empirical product-moments. In the case of the singlet correlations, the expectations are zero, hence the standard deviations are one and the means are zero; the physicist’s theoretical correlations are actually also the statistician’s theoretical correlations.

### 6.3. The International Journal of Theoretical Physics Paper, 2015

Christian (2015) [8] is a paper entitled "Macroscopic Observability of Spinorial Sign Changes under 2π Rotations" which appeared in the journal *IJTP*. I wrote to the editors, saying that the paper should be retracted because of its obvious errors, but the editors preferred to print my critique—for a fee. So this became a short paper Gill (2016) [29] entitled "Macroscopic Unobservability of Spinorial Sign Changes". It was accepted and published in *IJTP* on receipt of a hefty publication fee from me. Christian posted a rebuttal on arXiv but this was not published by *IJTP*; I do not know if he submitted it to that journal. He does incorporate his rebuttal in later papers, in which he deals systematically with all his critics. I did not look at it.

The paper describes a macroscopic “exploding balls” experiment in which we do measure the spins of various rotating hemispheres in various directions simultaneously. There is no quantum mechanics involved, no non-commutation. The experiment in question would definitely produce correlations which do not violate any Bell inequalities. The paper shows that the author has no mathematical understanding of Bell’s theorem at all. He also has no understanding of real Bell experiments, either. This is strange in the PhD student of Abner Shimony, who even met John Bell at international conferences on perhaps several occasions; once notably and surely memorably at Erice in Sicily.

### 6.4. The Annals of Physics Paper, 2015

Christian (2015) [9] is entitled "Local Causality in a Friedmann–Robertson–Walker Spacetime". This paper mysteriously vanished from the web-site of the once famous journal *Annals of Physics* after it had been accepted but before the next issue was definitive. Most of the editorial board have Nobel prizes. One guesses that they are not much involved in editorial processes. On the web page https://www.sciencedirect.com/science/article/pii/S0003491616300975 one can read
This article has been withdrawn at the request of the Editors. Soon after the publication of this paper was announced, several experts in the field contacted the Editors to report errors. After extensive review, the Editors unanimously concluded that the results are in obvious conflict with a proven scientific fact, i.e., violation of local realism that has been demonstrated not only theoretically but experimentally in recent experiments. On this basis, the Editors decided to withdraw the paper. As a consequence, pages 67–79 originally occupied by the withdrawn article are missing from the printed issue [vol. 373]. The publisher apologizes for any inconvenience this may cause.
Through an administrative error, the author had not been informed.

Amusingly, if there were a local realistic model of the singlet correlations, it would have been easy to experimentally violate Bell inequalities. The editors clearly have no clue whatsoever as to the logic of Bell’s theorem. The experiment would have proven that the “theoretical” result of Bell was actually wrong, just as Christian claims.

Quite amazingly, this paper included a computer simulation, allegedly of the author’s model, but actually a simulation of Pearle’s famous detection loophole model of 1970 [30]. The author does not himself read or write computer programs, but has some supporters who write software for him. Since it is impossible to implement his own model while still getting the right answer, they are forced to modify Christian’s own definitions or even to simply write code to implement already existing models earlier invented by others, of which there are many. It seems that one of them borrowed my code, written in early 2014 and published on internet and discussed in internet fora, without telling me or Christian. I have elsewhere [31] published further details about my re-discovery, correction, and implementation of Pearle’s model.

Incidentally, the Pearle detection loophole model, and the earlier discussed GA cancellation tricks, have nothing whatsoever to do with one another. Of course, the author does not even mention the mismatch. One marvels that referees can recommend publication of a series of papers with such evident self-contradictions, and that editors do not have a clue. I am getting old. O tempora, o mores!

### 6.5. The RSOS Paper, 2018

At last Christian managed to publish a really major paper [10] in the journal *Royal Society Open Science*, entitled "Quantum correlations are weaved by the spinors of the Euclidean primitives". Here he presents a (for him) new argument why Bell’s theorem is wrong. He claims that the use of expressions involving simultaneously measurement functions evaluated at several different measurement settings cannot possibly have any physical meaning and hence that proofs which depend on the evaluation of such expressions are physically meaningless and hence invalid.

But under local realism, the local hidden variable λ is supposed to exist, even if it cannot be observed directly, and various functions thereof are also supposed, by the assumption of local realism, to exist. Hence one can mathematically consider any combinations of those functions one likes, and if that leads to observable consequences, for instance, bounds on correlations which can be observed in experiments, then those bounds must be valid, if local realism is true.

Note that when I use the word “exists”, I speak as a mathematician, and I am discussing mathematical models. For me, local realism means the existence of a mathematical model which reproduces the predictions of quantum mechanics (or approximately does so). It is up to physicists and to philosophers to thrash out what they mean when they use words like local, real, or exist. The mathematics which they do should “stand alone”, the words used to describe various objects and to describe the relations between them are irrelevant.

Christian is not the first physicist to get confused on this issue. It keeps popping up. It has kept popping up for 50 years. Yes, the proof of the CHSH inequality involves combining measurement functions with different settings inserted but depending on the same value of the hidden value. Yes, quantum physics says that one cannot measure spin in different directions at the same time. So what? Who says we are doing that?

Apart from that, the paper yet again proposes a local hidden variables model; but a new one this time. The hidden variable λ is again a fair coin toss, taking the values ±1. It follows that the pair (A(a,λ),B(b,λ)) can only take the values (−1,−1), (−1,+1), (+1,−1) or (+1,+1). Their product can only take the values ±1 and the only probabilities around are $\frac{1}{2}$, $\frac{1}{2}$. This means that the expectation value of a product A(a,λ)B(b,λ), whatever the values of the settings *a*, *b*, can only be −1, 0 or +1. (I learnt this argument from Heine Rasmussen, it is beautiful).

The proof that the singlet correlations are produced by this model is more sophisticated than ever; and the paper actually claims to deal with all quantum correlations, not just the EPR-B correlations. The usual errors can be located in the proofs, together with a new one: a limit is computed as $s_{1}\to a$ and $s_{2}\to b$, while imposing the “physical” constraint $s_{1}=-s_{2}$. The notation is non-standard and not explained, but not surprisingly, this does not make much sense unless a=−b. The result is a correlation of ±1, which is indeed what is desired if a=−b, but not otherwise.

There is also a proof of an astonishing pure mathematical result in this paper, which has been separated off and submitted as a separate publication to a pure mathematics journal. I will deal with that at the end of this section. That paper has not yet appeared in print.

The referee reports are published online. Three referees wrote short reports saying that the paper was pure nonsense, giving some documentation and/or argumentation. Several earlier papers have been written and published in decent journals explaining why Christian’s model is no good, and the old criticism applies to the new paper, too. Two referees wrote very, very short reports saying no more and no less than that the paper was brilliant. The author has proudly remarked elsewhere that the journal kindly waived his publication fee, so at least that is something.

Again several folk wrote to the journal complaining about the paper and arguing that it should be withdrawn. I wrote complaining that my computer code had been stolen. A committee is evaluating the paper and an “expression of concern” has been posted. I have the impression that the journal is just waiting for the problem to go away.

### 6.6. The IEEE Access Paper, 2018

Again, a big success came with a publication in an *IEEE* journal, entitled “Bell’s Theorem Versus Local Realism in a Quaternionic Model of Physical Space” [11]. This is actually a revised version of the disappearing *Annals of Physics* paper “Local Causality in a Friedmann–Robertson–Walker Spacetime” which I discussed earlier. It combines elements from all previous works!

I complained of plagiarism to the journal, and I am told that an investigation is underway.

### 6.7. The Pure Mathematics Paper, 2019

In this latest paper "Eight-Dimensional Octonion-Like but Associative Normed Division Algebra" [12] which Christian says is submitted (to a pure mathematics journal), Christian claims he has found a counter-example to Hurwitz’s theorem that the only division algebras are R,C,H,O of dimensions 1, 2, 4, and 8. His “counter-example” is the real Clifford algebra $C\ell_{0,3}$ which is not a division algebra at all.

He writes “the corresponding algebraic representation space … is nothing but the eight-dimensional even sub-algebra of the $2^{4}=16$-dimensional Clifford algebra $C\ell_{4,0}$”.

This space is well known to be isomorphic to $C\ell_{0,3}$. The Wikipedia pages on Clifford algebras are very clear on this, and as far as I can see, well sourced and highly accurate. One can take as basis for the 8 dimensional real vector space $C\ell_{0,3}$ the scalar 1, three anti-commuting vectors $e_{i}$, three bivectors $v_{i}$, and the pseudo-scalar $M=e_{1}e_{2}e_{3}$. The algebra multiplication is associative and unitary (there exists a multiplicative unit, 1). The pseudo-scalar *M* squares to −1. Scalar and pseudo-scalar commute with everything. The three basis vectors $e_{i}$, by definition, square to −1. The three basis bivectors $v_{i}=Me_{i}$ square to +1.

Take any unit bivector *v*. It satisfies $v^{2}=1$ hence $v^{2}-1=\left(v-1\right)\left(v+1\right)=0$. If the space could be given a norm such that the norm of a product is the product of the norms, we would have ∥v−1∥.∥v+1∥=0 hence either ∥v−1∥=0 or ∥v+1∥=0 (or both), implying that either v−1=0 or v+1=0 (or both), implying that v=1 or v=−1, neither of which are true.

Recall that a normed division algebra is an algebra that is also a normed vector space and such that the norm of a product is the product of the norms; a division algebra is an algebra such that if a product of two elements equals zero, then at least one of the two elements concerned must be zero. A splendid resource on the Hurwitz theorem and the octonions is John Baez’s celebrated 2002 article in the *Bulletin of the American Mathematical Society* reproduced on his web pages http://math.ucr.edu/home/baez/octonions/. Full proofs are given but already the key definitions and main theorems are clearly stated in the first couple of pages.

To the computer-savvy person who is familiar with the basic concepts of geometric algebra, I recommend use of Dorst et al.’s [16] already mentioned program *GAviewer*, in order to verify for themselves whether or not the even sub-algebra of $C\ell_{4,0}$ is isomorphic to $C\ell_{0,3}$ and hence not a normed division algebra, in contradiction to the claims in all of Christian’s latest three papers, two published and one submitted.

The isomorphism can be verified by first drawing a correspondence between $\left(e_{1}\;e_{2}\right)$, $\left(e_{2}\;e_{3}\right)$, and $\left(e_{3}\;e_{1}\right)$ in the even sub-algebra of $C\ell_{4,0}$ with $e_{1}$, $e_{2}$, $e_{3}$ in $C\ell_{0,3}$. They anti-commute and square to minus one. The next step in establishing the isomorphism could be to verify that $\left(e_{1}\;e_{2}\;e_{3}\;e_{4}\right)$ in the even sub-algebra of $C\ell_{4,0}$ could correspond with $\left(e_{1}\;e_{2}\;e_{3}\right)$ in $C\ell_{0,3}$. To round things off one needs to find which remaining basis elements of $C\ell_{0,3}$ correspond to the remaining even basis elements of $C\ell_{4,0}$. A nice little exercise for the reader!

There do exist some exotic objects called “exotic spheres”: differentiable manifolds that are homeomorphic but not diffeomorphic to the standard Euclidean *n*-sphere. That is, a sphere from the point of view of all its topological properties, but carrying a smooth structure that is not the familiar one. They were first discovered by Milnor in 1956 for the 7-sphere, and have been further classified by him and many other researchers. It is unknown if exotic 4-spheres exist. There are no other exotic spheres in dimensions 1 up to 6; from then on, they only exist in some dimensions, and much is still unknown about them.

Christian claims that his model depends on the Hopf fibration, a wonderful way to look at the 3-sphere. He claims that a uniform distribution on $S^{3}$ relative to the Hopf fibration differs from the usual uniform distribution on $S^{3}$. But this makes no sense. “Uniform distributions” are defined relative to some symmetries, and the Hopf fibration does not introduce new symmetries to $S^{3}$ beyond the familiar ones of “flatland”, i.e., of Euclidean geometry.

## 7. Conclusions

The huge number of attempts (several per year, both by amateurs and by professionals) to refute Bell’s theorem has to do, in my somewhat biased opinion (I have a lot of interactions with “Bell deniers” who are physicists, but who may not be typical physicists), with two problems in physics. One is that physicists use mathematics as a language in which they speak about the physical world. This makes it harder for them to distinguish between models and reality. Mathematicians do not only calculate; they also study mathematical models, thought of as stand-alone abstract worlds. The common distinction between theoretical physics (part of physics) and mathematical physics (part of mathematics) corresponds, in my mind, to this same difference between on the one hand talking with mathematics about the real world and on the other hand talking with logic about imaginary or toy mathematical worlds. The second problem is the lack of appreciation of many old school physicists for modern probability theory, and lack of knowledge about modern statistics. Both of these problems are historical problems, science is evolving, and new generations see scientific hierarchies and divisions differently from old generations. Incidentally, Bell, who was a physicist, had a consummate understanding of statistical issues, though he was hampered by the language he was obliged to use in order to communicate with the physicists of his day.

Christian’s main problem is that he does not understand Bell’s theorem as a mathematical theorem, and that he does not have the mathematical discipline to be sufficiently careful with notation. On the other hand he is so confident of his physical insight that he is not going to let any mathematical difficulties get in his way. The ultimate authority for him is nature, not logic. However he does not provide any new understanding of nature.

His original errors were to ignore the fact that measurement outcomes in Bell’s theorem, and also in experimental tests of Bell’s theorem, are binary. The statistics which experimentalists calculate are called correlations, but they are just simple functions of counts of various kinds of combined events. Changing the co-domain of the measurement functions, by which Christian means that he embeds the numbers ±1 in a bigger space, does not change any of the many different proofs of Bell’s theorem, since whether we think of +1 and −1 as a quaternion, octonion, or whatever, adding them gives us integers; further arithmetic with these numbers does not take us beyond the rational numbers. The topology of physical space is totally irrelevant. Bell’s theorem depends on notions of locality and realism, and notions of causality connected to temporal and spatial separation. Bell’s theorem is not about angles or reference frames. This is a common misapprehension. Most of its proof is not about quantum mechanics. That is another common misapprehension.

In his very first paper, Christian took as measurement outcomes the bivectors of the standard Clifford algebra of real 3D space. Plenty of authorities pointed out that this does not disprove Bell’s theorem—it results in some mathematics which is irrelevant to Bell’s theorem (in particular, see a nice paper by Weatherall, published in *Foundations of Physics*). Clearly, Christian realised that he had made a mistake, though he never admitted to one. In his next try he did define measurement functions which took as outcomes the values ±1. He thought of them as points in Clifford algebra but now had to redefine correlation and hide a sign error in order to get the right answer. The sign error was introduced by defining two different products: the usual geometric product, and its transpose. His hidden variable was which product should be used. The result of all this is pure nonsense. If the hidden variable is binary then all correlations can only be zero, minus one, or plus one. There is no need to look for the errors in the derivation.

Later papers introduced more mathematical sophistication which in effect hid the same errors deeper in a fog of apparently sophisticated but actually naive mathematics. A novel limit operation was introduced. It became apparent that Christian sees the mathematical expression “limit as *s* tends to *a*” not as it is formally defined in calculus but as physical imagery, a particle with a property *s* moves to a location where there is a device with property *a*. This is a rather touching, even child-like, way to do physics; it is no way to do mathematics. Christian also came up with a new argument for the physical meaninglessness of the difference between correlations which is analysed in the CHSH in equality. He forgets that if the hidden variable λ is supposed to exist, and if some measurement functions are also supposed to exist, then one can look at combinations of those functions of that variable. The expressions one meets in a mathematical derivation need not have line-by-line physical interpretations. This misapprehension is widely spread. Bell’s argument is thought to be invalid because he combines mean values which can only be observed in different experiments.

I like to mention the following example. Suppose we are interested in a population of married couples. We take a sample of the men, and another sample of the women, and look at the difference between the two average heights. We will get a decent estimate of the average difference in height of a woman and her husband. Of course we would get an estimate with smaller variance if we took a sample of couples. In the quantum case, the couples cannot be sampled. The point of Bell’s theorem is that the couples could not exist. Physicists, like most ordinary folk, are often unconvinced by proofs by contradiction: it is strange to assume the opposite of what one wants to prove.

Finally, Christian believes that he has found a counter-example to the Hurwitz theorem, that the only division algebras are the real numbers, the complex numbers, the quaternions and the octonions. This result from abstract algebra is famous, and has numerous different proofs. Similarly, Bell’s theorem is famous, and has numerous different proofs. Bell’s theorem can be seen as a simple result in probability theory, as a simple theorem in functional analysis, as a simple theorem in approximation theory, as a simple theorem in combinatorics (discrete mathematics). The Hurwitz theorem is a good deal harder to prove, and a good deal deeper. Fortunately, there is no need to go into the mathematical details of Christian’s work to see that his conclusions must be wrong – the conceptual misunderstandings of Bell’s theorem are immediately obvious. His counter-examples must be wrong. His arguments that Bell made some mistakes are completely misguided.

Many attempts to disprove Bell’s theorem use very sophisticated mathematics to come up with a counter-example, the mathematical details of which are beyond most people’s capacity to check. In fact, the authors’ enthusiasm with their vindication of Einstein and their proof that Bell and Bohr were wrong also clouds their vision and the errors are hidden in little notational slip-ups at the end of a huge analysis, when they feel they are “on the home stretch”, and discipline is relaxed. In fact this is much like accidents in mountain climbing.

There is no need to analyse the proof, since one usually can easily see by the general introductory remarks in such papers that the authors do not understand the point or the logic of Bell’s theorem. I hope that readers of this cautionary tale will learn not to be impressed by apparently learned use of abstract mathematics, and understand how simple Bell’s theorem is. Extraordinary claims require an extraordinary degree of support. Usually it is not necessary to work through obscure mathematical details to find an error. One can see in the introductory words that the author did not understand Bell’s theorem or its proof in the first place, hence his or her counter-example or meta-physical arguments can reasonably be taken with a grain of salt. As a last resort, ask the author to establish their claim by publishing computer programs which can be run on a network of ordinary home computers and which violate Bell inequalities by a substantial margin (see Gill (2014) [24] for a discussion of the design of “quantum Randi challenges”, a concept invented for pedagogical and outreach purposes by Sascha Vongehr). If the Bell-denier succeeds, the whole world will be able to verify that they are right; they will get the Nobel prize for performing a reproducible physical experiment which disproves Bell’s theorem. More likely, they will not succeed, but perhaps they will learn from their attempts what they are up against.

We also learn that the peer-reviewed literature is full of pure nonsense. Even established and reputable publishing houses now offer their own “tame quality predatory journals” in order to satisfy the huge world-wide economic/social demand for peer-reviewed publishing venues. After all, how else can university managers decide who should get promotion or tenure? At least, these more established and more professional publishing houses can offer a bigger guarantee that their publications will still be accessible in, say, twenty years, and maybe even in a century (if technologically advanced humankind is still around). More than ever, young researchers have to learn not to trust anything, even if that is going to impede their own academic progress. The journals *IJTP* and *Annals of Physics* used to have good reputations. In their defence, it must be said that Christian himself deliberately formulated his work as an exercise in relativity theory in order to avoid close inspection by experts in quantum information. The journals *RSOS* and *IEEE Access* have a world expert on geometric algebra on their editorial boards. Of course, such a person would believe that GA should de-mystify Bell’s theorem and possibly then even solve the problems of interpretation of quantum mechanics, just as David Hestenes had intimated. Unfortunately such an expert is typically not an expert on Bell’s theorem, quantum information, and the interpretation of quantum mechanics.

A bit more amateur psychology: we all too easily see what we want to see. So, dear reader, do not trust me either. Check and find out for yourself whether or not I am right. You might even like to learn a bit of geometric algebra, like I had to. Maybe it will come in useful for you. I look forward to hearing from you exactly what I have got wrong.

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
