# Peer review of "Does Geometric Algebra Provide a Loophole to Bell’s Theorem?"

_entropy, 2019, doi:10.3390/e22010061_

Round 1
Reviewer 1 Report
It is an article of synthesis whose contribution of the author did not appear well. Accept after minor revision (corrections to minor methodological errors and text editing); try to clarify your contribution in a section named my results and reduce the summary of your article because it appears very long.
Author Response
Your main comment was "It is an article of synthesis whose contribution of the author did not appear well. Accept after minor revision (corrections to minor methodological errors and text editing); try to clarify your contribution in a section named my results and reduce the summary of your article because it appears very long".
I have thoroughly revised and restructured the whole article. I hope that my aims and contribution are now completely clear. The abstract is now short. The paper has grown somewhat in length because a number of references and discussion of some of those references needed to be added too.
Reviewer 2 Report
The author examines some papers and monograph by Joy Christian (2007,2011,2014), see [1,2,3], exposing a conceptual and an algebraic error.
Of concern is ,of course, Bell's Theorem.
However, it is to be notice that the arguments of the author do not seem so clear, too.
The indicated paper has some interest. The author indicates some not correct affirmations in various papers and monograph from Joy Christian (2007, 2011, 2014), precisely disproof of Bell's Theorem. However, such a not-correctness proof in its time does not appear so clear and convincing and it needs more detailed version.
Author Response
Your main criticism was "However, such a not-correctness proof in its time does not appear so clear and convincing and it needs more detailed version." I have expanded the not-correctness proofs of the important papers of Christian, so that, I hope, the central and fatal errors are clear. At the same time I have also drawn wider conclusions about similar attempts by numerous other authors to disprove Bell, so that I hope the paper now has wider and more constructive value.
Round 2
Reviewer 2 Report
This is a really strange paper!
In the second version the author changed very much the presentation.
He affirms that some papers and monograph from Joy Christian on the famous Bell's Theorems are not correct.
He provides a proof for this affirmation.
Indeed, this proof is not so simple as one would hope.
Some whole pages are of difficult reading and not clear.
page 9, 12a-18a are not absolutely clear.
p.10, 15-23a not clea.
p.11, 1-16a not clear.
p.13, 1-2a: not clear.
Many pages are absolutely not readible, in particular, page 14.
Also page 15 is incomprehensible.
Page 18 is not comprehensible, too.
page 19:It seems that part of the beginning in 7 is lacking.
I invite the author to submit a version improving the indicated points.